# Integrated Analysis of lncRNA–miRNA–mRNA Regulatory Network in Rapamycin-Induced Cardioprotection against Ischemia/Reperfusion Injury in Diabetic Rabbits

**DOI:** 10.3390/cells12242820

**Published:** 2023-12-12

**Authors:** Arun Samidurai, Amy L. Olex, Ramzi Ockaili, Donatas Kraskauskas, Sean K. Roh, Rakesh C. Kukreja, Anindita Das

**Affiliations:** 1Division of Cardiology, Pauley Heart Center, Internal Medicine, Virginia Commonwealth University, Richmond, VA 23298, USA; arun.samidurai@vcuhealth.org (A.S.); ramzi.ockaili@vcuhealth.org (R.O.); donatas.kraskauskas@vcuhealth.org (D.K.); kyungno1@gmail.com (S.K.R.); 2Wright Center for Clinical and Translational Research, Virginia Commonwealth University, Richmond, VA 23298, USA; alolex@vcu.edu

**Keywords:** diabetes, ischemia/reperfusion injury, long non-coding RNAs, microRNAs, mRNAs, mTOR, rapamycin, MALAT1, apoptosis

## Abstract

The inhibition of mammalian target of rapamycin (mTOR) with rapamycin (RAPA) provides protection against myocardial ischemia/reperfusion (I/R) injury in diabetes. Since interactions between transcripts, including long non-coding RNA (lncRNA), microRNA(miRNA) and mRNA, regulate the pathophysiology of disease, we performed unbiased miRarray profiling in the heart of diabetic rabbits following I/R injury with/without RAPA treatment to identify differentially expressed (DE) miRNAs and their predicted targets of lncRNAs/mRNAs. Results showed that among the total of 806 unique miRNAs targets, 194 miRNAs were DE after I/R in diabetic rabbits. Specifically, eight miRNAs, including miR-199a-5p, miR-154-5p, miR-543-3p, miR-379-3p, miR-379-5p, miR-299-5p, miR-140-3p, and miR-497-5p, were upregulated and 10 miRNAs, including miR-1-3p, miR-1b, miR-29b-3p, miR-29c-3p, miR-30e-3p, miR-133c, miR-196c-3p, miR-322-5p, miR-499-5p, and miR-672-5p, were significantly downregulated after I/R injury. Interestingly, RAPA treatment significantly reversed these changes in miRNAs. Gene Ontology (GO) and Kyoto Encyclopedia of Genes and Genomes (KEGG) enrichment analysis indicated the participation of miRNAs in the regulation of several signaling pathways related to I/R injury, including MAPK signaling and apoptosis. Furthermore, in diabetic hearts, the expression of lncRNAs, HOTAIR, and GAS5 were induced after I/R injury, but RAPA suppressed these lncRNAs. In contrast, MALAT1 was significantly reduced following I/R injury, with the increased expression of miR-199a-5p and suppression of its target, the anti-apoptotic protein Bcl-2. RAPA recovered MALAT1 expression with its sponging effect on miR-199-5p and restoration of Bcl-2 expression. The identification of novel targets from the transcriptome analysis in RAPA-treated diabetic hearts could potentially lead to the development of new therapeutic strategies for diabetic patients with myocardial infarction.

## 1. Introduction

Diabetes Mellitus (DM) is a major risk factor for myocardial infarction (MI), which augments the deleterious pathological outcomes following MI [1,2]. The mortality rate after an incident of MI is higher in diabetes patients compared to non-diabetic subjects [3,4,5]. In diabetic patients, hyperglycemia, insulin resistance with a compensatory hyperinsulinemia, excessive production of fatty acid, and imbalanced lipid metabolism lead to an increase in systemic oxidative stress, protein kinase C activation, and the production of advanced glycation product (AGE) [6,7]. These all promote endothelial cell apoptosis and impair endothelial dysfunction, which causes vascular inflammation and vasoconstriction [8]. Moreover, the higher tendency of coronary artery calcification (CAC) in diabetic patients, due to increased oxidative stress and inflammatory cytokine production, endothelial dysfunction, alteration in mineral metabolism, and release of osteoprogenitor cells from the marrow into the circulation, lead to an increase in arterial rigidity and atherosclerotic plaque burden [9]. The presence of diabetic neuropathy also enhances the incidence of death after an episode of MI [10] largely due to the lack of pain and attention while experiencing MI. The national diabetes statistics report in 2022 revealed that about 37.3 million people, i.e., 11.3% of the total US population, have been diagnosed with diabetes [11]. The prevalence of DM in the global population has increased exponentially from 108 million in the year 1980 to 422 million in 2014, and is expected to have an upward trajectory in the coming years. Age-adjusted diabetes rate is expected to increase by 59.7% between the years 2021 and 2050, resulting in a global population of 1·31 billion people living with diabetes in 2050 [12]. This alarming increase in the rate of diabetes is largely due to the prevalence of childhood obesity and the presence of gestational diabetes. Several factors increase the rate of diabetes including lack of exercise, lifestyle changes, awareness, diet [13], and demographic characteristics [14]. In fact, there was a 3% increase in the mortality rate due to diabetes between the years 2000 and 2019 (https://www.who.int/news-room/fact-sheets/detail/diabetes; accessed on 5 April 2023) and this number is projected to increase to 783 million people by the year 2045 (https://diabetesatlas.org/; accessed on 12 September 2023). Moreover, patients with diabetes are twice as likely to develop cardiovascular diseases, including MI [1,15], which significantly increases post-MI fatality rates in diabetic patients [16,17,18,19,20].

Diabetes changes the metabolic environment of cells and modifies the epigenetic pattern by regulating the molecular mechanism of gene expression, which disturbs the cellular signaling [21,22]. Differentially methylated regions (DMRs) in the whole genome, especially the pancreatic and duodenal homeobox 1 (PDX1), a key transcription factor present in the pancreatic islet cells that regulate the insulin secretion [23], have been reported to play an important role in diabetes. Diabetes also influences epigenetics and alters the expression of gene patterns. Human islets incubated in high glucose (19 mM) altered the methylation pattern of five key genes such as GLRA1, RASD1, VAC14, SLCO5A1, and CHRNA5 [24]. Gene enrichment also suggests that signaling pathways including TGF-beta, Notch, and SNARE interactions in vesicular transport are involved in the islet function and diabetes. These epigenetic alterations control the expression of mRNAs and non-coding RNAs (ncRNAs), such as microRNAs (miRNAs), long non-coding RNAs (lncRNAs), and circular RNAs (circRNAs) [25,26,27,28]. Among these ncRNAs, miRNAs [29] and lncRNAs are well studied and established to play an important role in gene regulation both during normal and pathological conditions [27,30,31,32]. miRNAs are small ncRNAs, typically ranging from 22 to 25 nucleotides (nt) in length, and are evolutionarily conserved across species, including humans, mice, rats, and rabbits [33,34,35,36,37,38,39,40]. In most cases, miRNAs interact with the 3′ UTR of target mRNAs and regulate their expression [41]. A partial similarity in the binding seed sequence results in the suppression of mRNA, whereas a complete complementarity in the seed region (7–8 mer) leads to the degradation of its target mRNA [42,43].

LncRNAs are a class of ncRNA transcripts that are longer than 200 nt bases and are characterized by their unique localization in the cell. Unlike miRNAs, lncRNAs are not conserved across species, but exhibit high tissue specificity. LncRNAs can be identified as ubiquitously expressed long non-coding RNAs (UE lncRNAs) from tissue-specific lncRNAs (TS lncRNAs) [44]. For example, the expression of lncRNA MIR17HG is specific to lungs, and plays an important role in cell survival, proliferation, differentiation, and angiogenesis [45]. The MIR17HG level is less abundant in non-small cell lung cancer and overexpression of MIR17HG reduces the methylation of the miR-142-3p gene to upregulate its expression, which inhibits cancer cell invasion and migration [46]. Several TS lncRNAs such as MYH2, TNNT2, and SLC8A1 are unique to heart tissue and have known functions in muscle contraction and relaxation, atrial cardiac muscle tissue morphogenesis, and cardiac muscle development. LncRNAs contain several characteristic features found in protein coding genes, such as the presence of 5′cap, exons and poly A tail, and post-transcriptional splicing activity. However, lncRNAs lack an open reading frame (ORF), which is essential for the synthesis of full-length functional protein. Recent reports also suggest that lncRNAs can translate to small peptides capable of executing important biological functions [47,48,49].

Both miRNAs and lncRNAs are critical for normal growth/organ development and play an important role during pathological situations. miRNAs and lncRNAs are involved in a variety of biological processes, including diabetes [50,51], ischemia/reperfusion (I/R) injury [52,53,54,55], and other cardiovascular diseases [31,56,57,58]. The specific interaction between miRNA and lncRNA during pathological conditions generates a fine line of gene regulation with much greater precision [59,60,61]. The three-way interaction between lncRNA, miRNA, and mRNA provides an additional layer of post-transcriptional regulation and results in unique epigenetic control, specific to tissue and cellular types. LncRNA and miRNA through their sponge mechanism regulate the expression of mRNA and dictate the DNA methylation and histone modification [62]. One of the classical examples for the three-way interaction between lncRNA, miRNA, and mRNA is evident from the report by Cheng et al., where lncRNA HOTAIR epigenetically suppressed the expression of miR-122 through DNA methyltransferases (DNMTs)-mediated DNA methylation via upregulation of Enhancer of zeste homolog 2 (EZH2) [63]. The suppression of miR-122 by HOTAIR through DNA methylation resulted in the activation of Cyclin G1 and enhanced tumor growth in hepatocellular carcinoma (HCC). The signature pattern of miRNAs can be regulated by small molecules or drugs used for therapeutic purpose, which alter the signaling pathways.

The mammalian target of rapamycin (mTOR), a nutrient-responsive serine-threonine kinase, is persistently active in the hearts of diabetic mice [64,65,66]. Treatment with mTOR inhibitor rapamycin (RAPA) improves cardiac function in diabetic mice by attenuating oxidative stress and altering contractile proteins [64]. We previously demonstrated cardioprotective effects of RAPA against I/R injury both in diabetic and non-diabetic mouse models [67,68,69]. Administration of RAPA at reperfusion provides protection against myocardial I/R injury by limiting infarct size, myocardial fibrosis, inflammation, and cardiomyocytes apoptosis in diabetic rabbits [52,70] and mouse models [69,71]. Treatment with RAPA following MI in diabetic subjects was shown to change the expression of several miRNAs and alter important signal cascades related to cardiomyocyte apoptosis and left ventricle dysfunction [57,72]. Alternatively, several miRNAs can also target components of mTOR complexes and act as endogenous inhibitors similar to small molecule drugs [73,74]. We showed that chronic RAPA treatment in diabetic mice attenuated MI via the STAT3–miR–17/20 signaling pathway [69], while treatment at reperfusion limited myocardial injury in diabetic rabbits by restoring AKT signaling through miR-302a [52].

To further obtain novel insights into the role of miRNAs in cardioprotection, we performed unbiased miRNA expression profiling from a global perspective and deduced DEmiRs (differentially expressed miRs) in the hearts subjected to I/R injury in diabetic rabbits. We analyzed the lncRNA–miRNA–mRNA paired interaction network and target prediction using a bioinformatic approach to identify DEmiRs. We also validated the expression of significantly altered selected miRNAs and lncRNAs by quantitative real-time PCR (qPCR) and confirmed the expression levels of their target protein.

## 2. Materials and Methods

### 2.1. Animal Care and Ethic Statement

New Zealand male rabbits (age: 3–4 months; body weight (BW): 2.8–3.0 kg) were purchased from Robinson Services Incorporated (RSI, Mocksville, NC, USA). All animal experiments were performed in accordance with USDA regulations and were approved by the Institutional Animal Care and Use Committee at the Virginia Commonwealth University. (IACUC protocol#AM10109; Preconditioning of the Ischemic Heart in Rabbits, approval date: 10 November 2021; Expiration date: 9 November 2024).

### 2.2. Rabbit Model of Type I Diabetes

The protocol for inducing Type I diabetes was previously described [52,75]. Briefly, New Zealand male rabbits (*n* = 9) were injected with alloxan monohydrate (125 mg/kg of BW, Sigma Aldrich, St. Louis, MO, USA) and confirmed for the development of DM. After establishing DM (blood glucose level > 220 mg/dL), animals were maintained for 4 weeks for further experiments.

### 2.3. Conscious Ischemia/Reperfusion Injury

Diabetic rabbits were randomized into three groups (*n* = 3/group): Control (DM), DM subjected to I/R injury (DM + I/R), and DM + I/R treated with RAPA (DM + I/R + RAPA). The rabbit model of conscious I/R has been described previously [52,75,76,77]. Briefly, a balloon occluder was implanted on the top of the left coronary artery in rabbits under anesthesia. After seven days, the sedated rabbits were subjected to a 45 min ischemia followed by 3 days of reperfusion by inflating/deflating the hydraulic balloon occluder. For the DM + I/R + RAPA group, animals were infused with either RAPA (0.25 mg/kg of BW; i.v.) or DMSO (vehicle) 5 min before the onset of reperfusion via marginal ear vein catheter. After 3 days of reperfusion, the rabbits were anesthetized by intramuscular administration of ketamine (35 mg/kg), Xylazine (5 mg/kg), and Atropine (5 mg/kg), and hearts were collected. Finally, rabbits were euthanized by administering saturated potassium chloride via ear auricular artery. Left ventricular (LV) tissues were collected and immediately frozen in liquid nitrogen and stored at −80 °C until further studies.

### 2.4. RNA Isolation and miRNA Array

Total RNA including small RNA, mRNA, and lncRNA was isolated from the risk area of LV tissue (3-days I/R) using miRNeasy mini kit (QIAGEN Sciences, MD, USA). The concentration and purity of the isolated RNA were verified using a Nanodrop ND-1000 spectrophotometer (Agilent Technologies, Santa Clara, CA, USA). The pooled RNA isolated from 3 different biological replicates from the 4 experimental groups was subjected to miRNA-Array analysis by loading on the microRNA array (miR-array) chip (LC Sciences Company, Houston, TX, USA). A detailed procedure of the miRNA-Array assay can be found in the publication by Zhou et al. [78]

Total RNA samples (1 µg) were 3′-extended with a poly(A) tail using poly(A) polymerase by incubating at 37 °C water bath for 15 min. An oligonucleotide tag was then ligated to the poly(A) tail for later fluorescent dye staining using T4 DNA Ligase (Invitrogen, Waltham, MA). Hybridization was performed on a µParaflo microfluidic chip at 40 °C for 16 h using a microcirculation pump (Atactic Technologies, Houston, Texas, USA) [78]. On the microfluidic chip, each detection probe consisted of a chemically modified nucleotide coding segment complementary to target miRNA (from miRBase, http://mirbase.org; accessed on 12 September 2023) or other RNA (control or customer defined sequences) and a spacer segment of polyethylene glycol to extend the coding segment away from the substrate. The detection probes were made by in situ synthesis using PGR (photogenerated reagent) chemistry. The hybridization melting temperatures were balanced by chemical modifications of the detection probes. After RNA hybridization, tag-conjugating Cy3 dye was circulated through the microfluidic chip for dye staining. Fluorescence images were collected using a laser scanner (GenePix 4000B Microarray Scanner, Molecular Devices, San Jose, CA, USA) and digitized using Array-Pro image analysis software (Media Cybernetics; version number is 4.0.0.41). Data were analyzed by first subtracting the background and then normalizing the signals using a LOWESS filter (Locally-weighted Regression) [79].

### 2.5. miRNA Normalization and Differential Expression Analysis

Normalization was performed by LC Sciences Company (Houston, TX, USA), who provided the background subtracted, planar corrected, LOWESS normalized raw data [78,79,80]. Differential expression analysis was also performed by LC Sciences using the probe duplicates on each chip as technical replicates to obtain *p*-values and fold changes for each miRNA. Expressions of miRNA were filtered for significance if their coefficient of variation was ≤0.5 and *p*-value ≤ 0.01.

### 2.6. Heat Map and Volcano Plot

Differentially regulated miRNAs between groups were shown as a heat map for visualization, which was done using R-Studio software, v4.1.3 [81]. Selected miRNAs with a significant expression change between groups are shown in a separate heat map where the red color indicates highly upregulated miRNAs and blue represents downregulated miRNAs. The volcano plot between the *p* value and fold change was generated to demonstrate the significance of the fold change in miRNAs between experimental groups using the R studio-Bioconductor package, where the red color indicated upregulated miRNAs, while the blue color denotes downregulated miRNAs. Black filled dots show differently expressed miRNAs with a non-significant *p* value.

### 2.7. miRNA–mRNA Target Prediction

The miRDB database (http://www.mirdb.org/index.html; accessed on 12 September 2023) custom target search tool was used to identify the predicted mRNA targets of each of the selected miRNAs by providing the miRNA sequences obtained from miRBase (https://mirbase.org/; accessed on 12 September 2023) to the miRDB custom prediction tool [63] (miRNA sequences used are available on GitHub at https://github.com/AmyOlex/Samidurai-Das_mTOR_miRNA_Cardioprotection; accessed on 12 September 2023). Targets for human, mouse, and rat interactions were searched for and placed in a spreadsheet for formatting and filtering. Interactions were filtered down to the top candidates with a Target Score ≥ 90, and formatted for input and visualized by Cytoscape v3.7. Second, the miRWalk [64] tool was used to identify predicted edges that were also present in miRDB (http://www.mirdb.org/index.html; accessed on 12 September 2023) and verified by MiRTarBase [65], a database of manually curated miRNA–mRNA interactions, to obtain high-confidence predictions that were present in multiple prediction tools. The 3′UTR, 5′UTR (Untranslated Region), and CDS (Coding Sequence) targets were obtained for miRNAs of interest from the DE analysis. Results were formatted for input to and visualization by Cytoscape v3.7 [66].

### 2.8. LncRNA–miRNA Target Prediction

The DIANA-LncBase v3 (https://diana.e-ce.uth.gr/lncbasev3/interactions; accessed on 12 September 2023) was used for prediction of lncRNA–miRNA interaction using the lncRNAs, MALAT1 (metastasis associated lung adenocarcinoma transcript 1), HOTAIR (HOX antisense intergenic RNA), GAS5 (growth arrest-specific 5), and MIAT (myocardial infarction associated transcript), and the following miRNAs of interest from the DE analysis: miR-1b, miR-29c, miR-193b-3p, miR-499-5p, miR-15b-5p, miR-140-3p, miR-199a-3p, miR-199a-5p, miR-214, and miR-320-3p. Human, mouse, and rat interactions were queried. Identified interactions were put into a spreadsheet and formatted for input into Cytoscape v3.7 for visualization of the network.

### 2.9. GO and KEGG miRNA Enrichment Analysis of DE miRNAs

Enrichment analysis using GO (Gene Ontology) terms [68,69] (http://geneontology.org/docs/go-enrichment-analysis/; accessed on 12 September 2023) and KEGG (Kyoto Encyclopedia of Genes and Genomes) (https://www.genome.jp/kegg/pathway.html; accessed on 12 September 2023) [70,71,72] pathways was performed by LC Sciences (Houston, TX, USA). The large filled circle equals 125 miRs and the smallest filled circle represents 25 miRs. 

### 2.10. Reverse Transcription and Real-Time PCR

The isolated RNA (10 ng) was subjected to reverse transcription reaction with an miRNA-specific RT-primer or with a random primer for lncRNA and mRNA expression using a microRNA reverse transcription kit (Thermo Fisher Scientific Inc., Waltham, MA, USA). Real-time PCR expressions of miRNA (miR-30a-5p, miR-30b-5p, miR-30c-5p, miR-30d-5p, miR-30e-5p, and miR-199a-5p) and lncRNA MALAT1, HOTAIR, and GAS5 were quantified using a TaqMan™ assay (Thermo Fisher Scientific Inc., Waltham, MA, USA). LncRNA and miRNA were normalized using U6 or Sno-202 small RNA, respectively. The lncRNA and miRNA expression levels are represented as fold changes using the Delta- Delta 2Ct method.

### 2.11. Protein Expression–Western Blot

Total soluble protein was extracted from the frozen LV tissues with lysis buffer (Cell Signaling, Danvers, MA, USA). Protein samples (50 μg) were resolved by SDS-PAGE, transferred to a nitrocellulose membrane, and blocked with 2.5% BSA (Bovine Serum Albumin, Cell Signaling, Danvers, MA, USA ). Membranes were incubated overnight with mouse monoclonal antibody specific for Bcl-2, Bax, and GAPDH (Cell Signaling, Danvers, MA, USA). The blots were then incubated for 1 h with anti-mouse secondary horseradish peroxidase-conjugated antibody (GE healthcare, Chicago, Illinois, USA) and developed using Western Lightning Plus–ECL substrate (PerkinElmer, Waltham, MA, USA). The densitometry analysis to quantitate the intensity of the protein band was performed using Image J software (ImageJ bundled with 64-bit Java 8, NIH, Bethesda, MD, USA).

### 2.12. Data Availability

The sequence data have been submitted to the BioSample database (hosted by the NCBI) (http://www.ncbi.nlm.nih.gov/biosample; accessed on 12 September 2023), under accession number PRJNA315318. R codes and related data/files for the study were deposited on GitHub at (https://github.com/AmyOlex/Samidurai-Das_mTOR_miRNA_Cardioprotection; accessed on 12 September 2023).

### 2.13. Statistical Analysi

All data were expressed as mean ± SEM. Analysis of variance tests or Student’s *t*-tests were used for statistical analyses. All miRNAs were considered to have significant differential expression if they were up- or downregulated by at least twofold. Statistical significance was determined when the *p*-value was less than 0.05 between groups.

## 3. Results

### 3.1. miRNA Array Analysis

Total RNA isolated from LV tissue of DM, DM + I/R, and DM + I/R + RAPA was subjected to miRNA array analysis (LC Sciences Company, Houston, TX, USA). A total of 806 unique customized miRNA targets (rat specific) were profiled (Figure 1A). Each chip contained four duplicate probes per miRNA and the results were used for DE analysis. Representative images of the microfluidic chip with fluorescent dye staining in DM, DM + I/R, and DM + I/R + RAPA groups are shown in Figure 1A. The heat map demonstrated differentially regulated miRs in diabetic rabbit hearts upon I/R injury and treatment with RAPA (Figure 1B).

### 3.2. Heat Map and Volcano Plot

A hierarchical cluster demonstrating a comprehensive overview of the DE miRNAs observed in DM, DM + I/R, and DM + I/R + RAPA groups is presented as a heat map for visualization (Figure 1B and Figure 2A) using R studio software v4.1.3. (Heatmap3 package). The heat map shows several miRs are differentially expressed in DM, DM + I/R, and DM + I/R + RAPA groups, which provides an overview of global changes in miRNA expression in the hearts of diabetic rabbits following I/R injury with/without RAPA treatment. The red color indicates a two-fold upregulation and the blue color represents two-fold downregulation of miRNAs between groups (Figure 2A). To delineate the significance of this DE miRNAs, volcano plots were constructed by the log2 of the fold change of miRs on the *x*-axis and negative log10 of their corresponding *p*-value on the *y*-axis between groups. The red color indicated upregulated miRNAs, while the blue color denoted downregulated miRNAs. Black filled dots show differently expressed miRNAs with a non-significant *p* value. (Figure 2B–D). A total of 194 variables were included in the analysis for comparison between the DM and DM + I/R groups (Figure 2B). Notably, miR-21-5p, miR-149-3p, miR-199a-5p, miR-365-5p, miR-214-3p, and miR-762 were upregulated less than two-fold, but miR-762 was upregulated by two-fold in the DM + I/R group as compared to the DM group. In addition, miR-194-5p, miR-378a-5p, miR-499-5p, miR-378a-3p, miR-378a-5p, miR-378b, miR-301a-3p, miR-30b-5p, miR-20b-5p, miR-29c-3p, and miR-29c-5p were downregulated by two-fold following I/R (DM + I/R) (Figure 2B).

A total of 243 miRNAs were regulated upon RAPA treatment (DM + I/R + RAPA) as compared to the DM + I/R group. Among these - DE miRNAs, miR-1-3p, miR-29b-3p, miR-29c-3p, miR-499-5p, miR-365-3p, miR-1956-5p, miR-30e-3p, miR-1b, miR-322-5p, miR-196c-3p, miR-352, miR-494-3p, and miR-133c were significantly induced by more than two-fold (Figure 2C). Furthermore, miR-672-5p, miR-539-5p, miR-154-5p, miR-543-3p, miR-379-3p, miR-379-5p, miR-299a-5p, miR-140-3p, miR-140-5p miR-497-5p, miR-379-3p, miR-411-3p, miR-199a-5p, miR-34a-5p, and miR-495 were downregulated by two-fold upon RAPA treatment (DM + I/R + RAPA) as compared to the DM + I/R group (Figure 2C).

Figure 2D shows that 196 miRNAs were differentially regulated between the DM + I/R + RAPA group and DM alone. RAPA treatment increased the expression of several miRNAs by more than two-fold in post-I/R myocardium (DM + I/R + RAPA) as compared to DM (Figure 2D), which were downregulated upon I/R injury in the myocardium (DM + I/R) as compared to DM (Figure 2A,C). The following miRNAs were increased by more than two-fold in post-I/R hearts after RAPA treatment: miR-1-3p, miR-223-3p, miR-494-3p, miR-5132-5p, miR-483-5p, miR-149-3p, miR-352, miR-365-5p, miR-762, miR-21-5p, miR-26b-5p, and miRs-let-7d-f-5p. The downregulated miRNAs in RAPA-treated post-I/R diabetic hearts were miR-24-1-5p, miR-18a-5p, miR-301a-3p, miR-194-5p, miR-497-5p, miR-378a-5p, and miR-221-5p.

### 3.3. Rapamycin Alters the Expression of miRNAs

Several unique miRNAs were differentially regulated in both the DM + I/R and in DM + I/R + RAPA treatment groups compared to the untreated DM control (Figure 3A,B). A total of 23 DE miRNAs were both upregulated and downregulated upon RAPA treatment as compared to the DM + I/R group. These were identified according to the criterion of adjusted *p*-value < 0.05 and absolute fold change ≥ 2. Notably, the expression of 10 miRNAs were suppressed following I/R injury in the diabetic rabbits (DM + I/R) as compared to the control group (DM) (Figure 3A). miR-1-3p, miR-1b, miR-29b-3p, miR-29c-3p, miR-30e-3p, miR-133c, miR-196c-3p, miR-322-5p, miR-499-5p, and miR-672-5p were restored after RAPA treatment (DM + I/R + RAPA) (Figure 3A). However, miR-365-5p, miR-365-3p, miR-352, miR-494-3p, and miR-1956-5p were upregulated after I/R injury (DM + I/R group), which was further induced with RAPA treatment (DM + I/R + RAPA group) (Figure 3B). Moreover, miR-199a-5p, miR-154-5p, miR-543-3p, miR-379-5p, miR-379-3p, miR-299a-5p, miR-140-3p, and miR-497-5p were induced following I/R injury, and RAPA treatment suppressed their expression (Figure 3C). Uniquely, the miR-30 family members were reduced in the DM/I/R group, which was restored upon treatment with the RAPA group (DM + I/R + RAPA) (Figure 3D).

### 3.4. Enrichment Analysis of miRNAs

To gain further insights into the biological pathway-specific function of DEmiRNAs, GO and KEGG pathway enrichment analysis was performed (Figure 4A,B). Nearly 125 miRNAs were significantly involved in the actin cytoskeleton pathway (<0.001). GO enrichment analysis based on cell function showed more than 300 miRNAs were involved in ATP binding (*p* < 0.0001; Figure 4A). More than 200 miRNAs were involved in calcium ion binding (*p* < 0.0002), which is of high importance to cardiac function. Interestingly, 100 miRNAs related to platelet activation were also enriched in the analysis (Figure 4A). Interestingly, more than 100 miRNAs were also involved in MAPK and cancer signaling pathways (Figure 4B). Notably, more than 25 miRNAs were associated with the apoptosis mechanism (*p* < 0.002). In addition, approximately 75 miRNAs were relevant to calcium signaling pathways, which are important for excitation-contraction coupling in cardiomyocytes. miRNAs that are involved in cell cycle regulation (~50 miRNAs; *p* < 0.002), which also play an important role in cardiac hypertrophy or dilated cardiomyopathy, were widely distributed in the pathway analysis (Figure 4B). Enrichment analysis also revealed miRNAs involved in several important signaling pathways including NOTCH, ErbB, lysosome, leukocyte trans-endothelial migration, cytokine-cytokine receptor interaction, ABC transporters, phosphatidylinositol, growth factor activity, regulation of actin cytoskeleton, extracellular space, and cell–cell signaling pathways (Figure 4A,B).

### 3.5. Target Prediction of miRNAs

The nucleotide sequences of miRNAs are highly conserved and show 100% similarity among rats, mice, humans, and rabbits [82]. Therefore, we used the miRNA repository of rats, mice, and humans to predict the targets. The highest number of targets were obtained against the human database compared to mice and rats (Figure 5A–C). Target prediction using miRWalk suggests that miR-199a can bind to Runt-related transcription factor-1 (RUNX1), a tumor suppressor, Histone-lysine N-methyltransferase 2A (KMT2a), a histone methyltransferase that positively regulates global gene transcription, and CD151-integrin complex, which permits the remodeling of epithelial cell interactions with the extracellular matrix and cell migration (Figure 5B). CD151 plays a critical role in tumor cell responses to laminin-5 and reveals the promotion of integrin recycling [83,84]. Also, miR-99-5p targets FZD8, which is known to reduce prostate cancer cell migration and invasion [85,86]. Gene target prediction also suggests that miR-214-5p can bind to Protein Phosphatase Targeting COQ7 (PPTC7) [87,88], Cytosolic Arginine Sensor for mTORC1 Subunit 2 (CASTOR2), and Anaphase Promoting Complex Subunit 16 (ANAPC16). The results also show that miR-320-3p can target SPRY4 (Sprouty RTK Signaling Antagonist 4), insulin-like growth factor (IGF1), Homolog of Drosophila Bicaudal D Cargo Adaptor 2 (BICD2), and Insulin Like Growth Factor 2 mRNA Binding Protein 3 (IGF2bp3) [89]. Interestingly miR-214-3p, which was decreased upon RAPA treatment in DM + I/R, was predicted to target Autophagy Related 12 (ATG12), Myocyte Enhancer Factor 2C (MEF2C), PTEN, PPARGC1B, and MAPK1. Moreover, miR-15b-5p was induced following I/R injury and significantly downregulated by RAPA. miR-15b-5p targets SMAD Family Member 7 (SMAD7), Notch Receptor 2 (NOTCH2) [90], an important player in the regeneration and cardiac repair post myocardial infarction, and Autophagy Related 9A (ATG9A). miR-15b-5p and miR-193b-3p can also target transforming growth factor-beta receptor 3 (TGFBR3). Importantly, miR-193b-3p was predicted to target TSC Complex Subunit 1 (TSC1), a key regulator of mTOR signaling.

### 3.6. Analysis of lncRNA–miRNA Network

Network analysis of lncRNA–miRNA interaction predicted that MALAT1 can sponge several miRNAs and regulate its target mRNAs (Figure 5D). Furthermore, bioinformatic evidence also suggests that MALAT1 can interact with miR-199a, miR-140-3p, miR-193b-3p, miR-320-3p, and miR-15-5p (Figure 5E). The interaction prediction also showed that lncRNA MIAT can interact with miR-320-3p and miR-15b-5p, and can have a sponge effect to regulate its mRNA targets (Figure 5D).

### 3.7. Expression of lncRNAs and miRNA by RT-QPCR

Based on the computational prediction, the expressions of lncRNAs, such as HOTAIR, GAS5, and MALAT1, were analyzed by RT-QPCR (Figure 6A–C). Results showed that lncRNAs, such as HOTAIR and GAS5, were induced upon I/R in diabetic hearts and treatment with RAPA at reperfusion suppressed the expression of these lncRNAs (Figure 6A,B). In contrast, MALAT1 was significantly reduced following I/R injury, and was recovered in the RAPA-treated group (Figure 6C). Bioinformatic-based analysis using Targetscan (www.targetscan.org; accessed on 12 September 2023) predicted that MALAT1 regulates Bcl-2 expression via sponging with miR-199a (Figure 6D). The downregulation of MALAT1 following I/R injury was associated with the increased expression of miR-199a-5p (Figure 6E). Treatment with RAPA suppressed miR-199a-5p expression concomitant to recovery of MALAT1, which apparently caused a sponging effect on miR-199a-5p (Figure 6D). The expression data of HOTAIR, GAS4, MALAT1, and miR-199a were normalized using the expression of U6 control (Figure 6A,B,C,E).

### 3.8. Protein Expression of Bcl2

Based on the bioinformatics evidence, we identified Bcl-2 as a potential target of miR-199a-5p (Figure 6D), which was confirmed by the suppression of Bcl-2 protein expression after I/R injury and the stimulation of miR-199a-5p (Figure 7A). Treatment with RAPA restored the expression of Bcl-2 and decreased the expression of miR-199a. The Bcl-2 protein expression was inversely related to the level of miR-199a expression in both DM + I/R vs. DM + I/R + RAPA groups (Figure 6E and Figure 7). Since the Bcl-2/Bax ratio is a marker for apoptosis, we quantified the expression of pro-apoptotic protein Bax in the treatment groups. As shown in Figure 7, the Bcl-2/Bax ratio was increased with RAPA treatment (*p*-value < 0.05; DM + I/R vs. DM + I/R-RAPA).

## 4. Discussion

Non-coding RNAs (ncRNAs), such as miRNAs and lncRNAs, are a crucial part of the transcriptome that act as post-transcriptional regulators of gene expression [91,92,93,94]. Initially labeled as “junk”, ncRNAs account for about 98% of all transcriptional output in the human genome and only the remaining 2% codes for functional protein [95,96]. miRNAs bind to either 3′UTR or 5′UTR of mRNAs based on their complementarity in their seed nucleotide sequence and suppress the translation process or degrade their target mRNAs [41,42,43]. More than 60% of the protein coding genes are regulated by miRNAs in the human genome [40,43]. LncRNAs, another class of ncRNAs, are also an effective regulator of gene expression. However, unlike miRNA, lncRNAs play a diverse role in the regulation of gene expression by interacting not only with RNA, but also with DNA [97,98] and proteins [99,100]. Although proteins are the ultimate functional unit that execute the cellular function, RNAs are important molecules that can shape and resolve the complexity associated with the gene expression and its regulation. The distinguished pattern in the expression of ncRNAs, which includes miRNAs and lncRNAs, is crucial in diabetes and MI. Diabetes is a metabolic disease, which alters the basic nature of cellular function and gene expression in response to injury, such as MI [101]. Therefore, it is critical to identify the differentiated signature pattern of miRNA expression during diabetes and upon I/R injury in diabetes. Similarly, it is also of importance to understand the effect of small molecule inhibitors, such as RAPA, in altering the miRNA expression. RAPA was demonstrated to restrict the damage inflicted on cardiomyocyte upon I/R injury in diabetes [52,71]. The identification of novel miRNAs regulated upon RAPA treatment may be useful in deciphering the mechanism of RAPA-induced cardioprotection and identifying potential novel targets of RNA- based therapy for heart diseases, including MI under diabetic conditions. Therefore, in this study, we conducted an unbiased global microarray-based miRNA expression analysis in heart tissue samples from diabetic rabbits following I/R injury with/without RAPA treatment. We also investigated the expression of lncRNAs in these groups and built an integrated network mapping between lncRNA, miRNA, and mRNA using bioinformatics tools. Computational prediction-based targets of DE miRNAs were deduced and GO and KEGG enrichment pathway analysis was performed.

Previous studies have established reciprocal regulatory effects of mTOR and miRNAs [102]. miRNAs affect mTOR by targeting its upstream regulators, including IGF-R, PI3K, PTEN, and AKT, or directly targeting different components of the mTOR. Conversely, mTOR regulates diverse cellular processes by suppressing or inducing multiple miRNAs, which causes several pathological conditions [102]. In this context, we showed that RAPA protects diabetic rabbit hearts against I/R injury by reducing apoptosis through the miR-302a-PTEN-AKT pathway [52]. mTORC1 was activated (indicated by phosphorylation of ribosomal S6), whereas mTORC2 activation (indicated by phosphorylation of AKT) and miR-302a were reduced with concomitant induction of PTEN (the target of miR-302a) following I/R injury in these hearts. RAPA treatment at reperfusion suppressed mTORC1 activity and PTEN, with restoration of miR-302a level and mTORC2 activity.

In the present study, we demonstrated a global change in the miRNA profile with RAPA treatment. We probed for a total of 806 miRNA targets that are well annotated and have high homology to the rabbit genome. A total of 243 miRNAs were DE between DM +I/R and DM + I/R + RAPA (greater than two-fold). The expression level of miR-1-3p was induced by RAPA treatment by more than six-fold, which was reduced upon I/R injury. The beneficial role of miR-1 was revealed in multiple studies. The level of miR-1-3p is suppressed in different biological sources of patients with T2D [103,104]. Targeted delivery of miR-1 in the cardiac tissue was shown to reverse pressure overload-induced cardiac hypertrophy and remodeling [105]. AAV9-mediated overexpression of miR-1 reduced several markers of cardiac hypertrophy including MYH6 and blunted adverse cardiac remodeling through maintenance of calcium homeostasis regulation by sarco/endoplasmic reticulum Ca^2+^ (Serca) [105]. Moreover, miR-1 overexpression also inhibited cardiac fibrosis and apoptosis by reducing the mRNA expression of transforming growth factor beta-1 (Tgfb1) and increasing protein expression of Bcl-2, while reducing the level of Bax, thereby altering the Bcl-2/Bax ratio [105].

Our results showed that miR-29b-3p is suppressed following I/R (DM + I/R group), while RAPA treatment significantly enhanced the expression of miR-29b-3p in the DM + I/R + RAPA group. It has been shown that the induction of miR-29b-3p attenuates cardiac fibrosis by targeting FBJ murine osteosarcoma (FOS) and the inactivation of the TGF-β pathway [106]. Pretreatment of cardiac fibroblast cells with miR-29b-3p mimics abrogated TGF-β1-induced fibroblast activation, cell proliferation, and reduced the expression of collagen I, collagen III, α-SMA, MMP2, and MMP9 protein [106]. miR-29b can regulate remodeling and stiffening of the aortic valve in db/db diabetic mice [107]. The reduction in the level of miR-29b was attributed to the abundance of expression of its target genes *Col1a1* and *Mmp2*. Interestingly, high glucose levels decreased the expression of miR-29b in human aortic smooth muscle cells. Several other studies also demonstrated the therapeutic effect of miR-29b in diabetes, cardiac fibrosis, apoptosis [108], and MI [109,110]. miR-29b KO mice develop vascular remodeling and systemic hypertension and heart failure with preserved ejection fraction (HFpEF) [111]. In addition, miR-29b targets PGC1α and regulates mitochondrial biogenesis [110]. Systemic delivery of low-dose miR-29b in diabetic mice improved myocardial performance with attenuation of cardiac fibrosis and improvement of endothelial function by suppressing the extracellular matrix gene expression [112]. Our recent study revealed that RAPA protected post-MI diabetic rabbit hearts by inhibiting myocardial remodeling and inflammation [70].

Our results also showed an increased expression of miR-499-5p in the RAPA-treated group, which may have an important role in protecting cardiomyocyte against I/R injury [113]. AgomiR overexpression of miR-499-5p in the acute MI model limited infarct size and reduced cardiomyocyte apoptosis through targeting programmed cell death 4 (PDCD4) [113]. Other studies also support the notion that miR-499 targets PDCD4 and strongly suggest that the overexpression of miR-499 is beneficial and cardioprotective, possibly through binding to SOX6 [114], thereby reducing cardiomyocyte apoptosis [115,116,117]. (Ref. [114]) A previous study also demonstrated that the expression levels of miR-1, miR-499, miR-133a, and miR-133b were depressed in the diabetic cardiomyocytes and streptozotocin-treated diabetic rat myocardium, which were restored by antioxidant N-acetylcysteine (NAC) treatment [118].

Our miRNA-array data also showed a robust 16-fold increase in the expression of miR-365-3p and a 14-fold increase in miR-365-5p in DM + I/R + RAPA compared to DM alone. It has been shown that the overexpression of miR-365-3p attenuated sepsis-mediated MI and inhibited the activation of the NF-κB inflammatory pathway through targeting myeloid differentiation factor 88 (MyD88) [119]. Rats injected with miR-365-3p mimic reduced inflammation and protected the heart from LPS-induced sepsis and acute MI [119]. Targeted delivery of RAPA via biomimetic nanoparticles for 7 days decreased macrophage proliferation in the aorta with the reduction of key pro-inflammatory cytokines in a murine model of atherosclerosis [120]. We predict that miR-365-3p may reduce the myocardial inflammation following MI in RAPA-treated diabetic rabbits.

Our miRNA array identified the induction of several miRNAs upon treatment with RAPA, which includes miR-196c-3p [121], miR-352, miR-494-3p [122,123], miR-29c-3p, miR-1956-5p, miR-133c [124], and miR-30e-3p [125,126], which have a critical role in cardioprotection. The results also showed upregulation of miR-154-5p and miR-543-3p after I/R injury. The overexpression of miR-30e-3p was shown to reduce LDH, IL-18, and IL-1β secretion and attenuate coronary microembolism (CME)-induced cardiomyocyte pyroptosis and inflammation [127]. Moreover, injection of AAV-miR-30e-3p in CME- induced MI in a rat model limited infarct size, improved cardiac function, and reduced the serum level of cTnI, IL-18, and IL-1β [127]. HDAC2, a target of miR-30e-3p, was upregulated in CME, which was reduced upon treatment with AAV-miR-30e-3p. Reports also suggested that miR30e attenuates isoproterenol (ISO)-induced cardiac remodeling and fibrosis [126] through targeting SNAI1 (Snail Family Transcriptional Repressor 1). Intramyocardial injection of miR-322-5p mimic reduced cardiac apoptosis and limited infarct size by about 40% through decreasing the expression of F-box and WD repeat domain-containing 7 (FBXW7) and regulating the Notch pathway [128]. It has been shown that the inhibition of miR-154-3p protects against cardiac dysfunction and remodeling [129]. Similarly, the inhibition of miR-543 reduced inflammation and provided protection against viral myocarditis through upregulation of sirtuin-1 (SIRT1) [130]. Interestingly, in the present study, we noticed that the post-I/R induction of both miRNAs (miR-154-5p and miR-543-3p) was significantly reduced upon RAPA treatment in post-I/R diabetic rabbits, which may provide cardioprotection in diabetic hearts against I/R injury.

We also observed an increase in miR-140-3p following I/R injury, which was suppressed after RAPA treatment. miR-140 expression has been shown to increase in the hypertrophic right ventricle of the Sugen5416-hypoxia-induced model of pulmonary arterial hypertension [131]. Pretreatment with anti-miR-140-3p is beneficial against afterload enhancement (AE) in Engineered heart tissue (EHT) and improved contractile force [132]. miR-140 directly targets mitofusin-1 (MFN1) and induces cardiomyocyte cell death. Interestingly, expression levels of miR-140 positively correlated with the elevated right ventricular systolic pressure (RVSP) [131]. Our study also showed that RAPA suppressed the expression of miR-497-5p in post-I/R. It has been shown that the inhibition of miR-497 is protective against MI [133]. miR-497 targets Wingless-Type MMTV Integration Site Family, Member 3A (WNT3A) and its inhibition using miR-497 antagomir contributes to bone marrow mesenchymal stem cells transplantation for MI treatment through regulation of the Wnt/β-catenin pathway [133].

As a master regulator, mTOR plays an important role in cardioprotection, diabetes and cellular metabolism, apoptosis, autophagy, mitochondrial biogenesis, nutrient sensing, and cardiac aging [52,69,71,134,135]. Gao et al. showed mTOR inhibition with RAPA reduced cardiomyocyte apoptosis following MI stress [136]. Further, RAPA also inhibited the expression of cleaved caspase-3, promoted cardiomyocyte autophagy, and alleviated ER stress through suppression of 78-kDa glucose-regulated protein (GRP78). Importantly, hyper-activation of mTOR signaling accelerates the cardiac aging process by compromising autophagy and the inhibition of mTOR activity rescued impaired autophagy in the heart, which delayed cardiac aging [135]. Treatment with RAPA improved contractile function and blocked the activation of hypertrophic genes in aged mouse hearts. Moreover, RAPA also inhibits inflammation associated with aging [137,138]. miRs can inhibit several components of the mTOR complex and mimic or even have additive effects similar to RAPA [57]. Activation of mTOR induces hypertrophy through miR-199a by targeting glycogen synthase kinase 3β (GSK3β)/mTOR complex signaling [139]. GSK3β blunts TSC1 and TSC2, the important negative regulators of mTOR signaling. The cardiomyocyte-specific overexpression of miR-199a induces cardiac hypertrophy and dysfunction [139]. Furthermore, miR-199a was shown to inhibit cardiac autophagy and activate mTOR signaling. Notably, treatment with RAPA restored cardiac autophagy and attenuated cardiac hypertrophy in miR-199a transgenic mice [139]. Furthermore, miR-199a is upregulated in tissues of cyanotic congenital heart defect patients [140]. In human cardiomyocytes, miR-199a was induced with hypoxia/reoxygenation and treatment with miR-199a mimic decreased the expression of glucose-regulated protein, GRP78 and activated transcription factor 6, the potential targets of miR-199a. The inhibition of miR-199a reduced endoplasmic reticulum stress and protected cardiomyocytes through the regulation of the myocardial unfolded protein response [140]. miR-199-5p was highly upregulated in the heart tissue of end-stage heart failure subjects [141]. The overexpression of miR-199-5p increased cardiomyocyte apoptosis through targeting Jun B proto-oncogene (JunB) [141]. Relevant to this observation, we found that miR-199a is upregulated in hearts of the DM + I/R group and treatment with RAPA decreased the expression of miR-199a. Our results suggested that Bcl-2 is a target of miR-199a, which plays an important role in the protection of cardiomyocytes against apoptosis. The miR-199a-5p was significantly reduced in the DM + I/R + RAPA group compared to the DM + I/R group alone.

LncRNA, a class of ncRNAs along with miRNA, adds an additional layer of gene expression and regulation. Gene expression is controlled with high precision and involves a complex network and interaction between various epigenetic factors including a three-way regulation between lncRNA, miRNA, and mRNA [18]. LncRNAs can act as an miRNA sponge and blunt the regulation of mRNA. Metastasis Associated Lung Adenocarcinoma Transcript-1(MALAT1) is one of the abundantly distributed lncRNA in several tissues and is often found elevated in various cancers [142] including metastasis lung cancer [143], breast cancer [144], and metastasis non-small cell lung cancer [145]. However, the role of MALAT1 in cardiovascular disease is understudied and reports suggest it may be beneficial against cellular apoptosis [146]. Our results showed that MALAT1 can interact with miR-155 and enhance the expression of Suppressor of Cytokine Signaling 1. The downregulation of MALAT1 induced cytokine release and apoptosis in Human Coronary Artery Endothelial Cells treated with ox-LDL. The upregulation of MALAT1 alleviated apoptosis and inflammation and reduced atherosclerosis [146]. Interestingly, the genetic ablation of MALAT1 inhibited the proliferation of endothelial cells and reduced neonatal retina vascularization [147]. Moreover, the pharmacological inhibition of MALAT1 using GapmeRs impeded blood flow recovery and capillary density induced by hindlimb ischemia [147]. MALAT1 was also shown to protect cardiomyocyte from ISO-induced apoptosis by sponging with miR-558 and via promoting ULK1-dependent autophagy [148]. Our network analysis of lncRNA–miRNA–mRNA predicted that MALAT1 contains a binding sequence for miR-199, which regulates the expression of Bcl-2, a target of miR-199. The MALAT1 and Bcl-2 expression were reduced following I/R injury with concomitant induction of miR-199a-5p. RAPA restored the expression of MALAT1 with downregulation of miR-199 and a concomitant increase of Bcl-2 expression. These studies suggest that MALAT1 can play a critical role in the reduction of apoptosis after MI under diabetic conditions.

Similarly, HOX antisense intergenic RNA (HOTAIR) plays an important role in cell differentiation and regulation of chromatin dynamics to control the gene expression [149]. HOTAIR is often found upregulated in several cancers including breast cancer [150,151,152], prostate cancer [153], and pancreatic tumors [154]. However, the role of HOTAIR in MI is still evolving and a recent finding demonstrates that HOTAIR induces myocardial fibrosis through competitive binding with miR-124 [155,156]. The inhibition of HOTAIR using lenti-shHOTAIR blunted myocardial fibrosis induced by Ang-II treatment in mice [156]. Further, knockdown of HOTAIR reduced the expression of Wnt5a and altered ERK and JNK signaling in atrial fibroblasts [156]. HOTAIR was also shown to promote myocardial fibrosis through Wnt signaling [157]. HOTAIR directly interacted and upregulated the expression of Unconventional prefoldin RPB5 interactor, also called (URI1), and overexpression of HOTAIR upregulated several cardiac fibrosis-related genes, including Collagen I, Collagen III, and α-SMA [157]. Relevant to this observation, it has been demonstrated that HOTAIR was upregulated during I/R injury in mice [158], which was associated with the downregulation of the interacting partner, miR-126, most prominently at 12 h as compared to 6 h of I/R injury [158]. Moreover, the inhibition of HOTAIR abolished H_2_O_2_-stimulated induction of LDH release and caspase-3 activity in H9c2 cardiomyocytes cells [158]. This study also identified Serine/arginine-rich splicing factor 1 (SRSF1) as a target of miR-126, and a network loop between HOTAIR-miR-126 and SRSF1 influenced I/R injury [158]. HOTAIR also plays an important role in the induction of inflammation and apoptosis in cardiomyocytes [159]. The HOTAIR and tumor suppressor PDCD4 are upregulated in the LPS-induced sepsis model in mice. PDCD4 promotes sepsis through activation of the NF-κB pathway and inhibition of IL-10 production. Interestingly, the inhibition of HOTAIR using AAV9-sh-HOTAIR improved the ejection fraction that was impaired upon LPS treatment and attenuated apoptosis [159]. Also, knockdown of HOTAIR reduced the expression of TNF-α, IL-6, and IL-1β in both LPS-treated mice and H9C2 cells [159]. Unlike a classical pathway, where lncRNA sponges with miRNA and regulates the target mRNA, HOTAIR increased the expression of PDCD4 through recruitment of Lin28, which in turn interfered with the maturation of let-7 miRNA and controlled the miRNA synthesis [159]. However, other studies suggest HOTAIR negatively regulates cardiac myocyte apoptosis in myocardial I/R injury [160]. Cardiac-specific overexpression of HOTAIR using the AAV9 vector limited infarct size and prevented cardiac myocyte apoptosis through activation of AMPKα and competitive binding of miR-451 [160]. Our results showed that the expression of HOTAIR was elevated following I/R injury and treatment with RAPA significantly reduced HOTAIR levels.

The lncRNA Growth arrest-specific 5 (GAS5) regulates several important biological processes, including cell growth arrest, cell proliferation, and apoptosis [161,162,163]. GAS5 was shown to influence myocardial I/R injury through molecular sponging with miR-532-5p and regulating PTEN-AKT signaling [164]. GAS5 was significantly upregulated following MI in rats and its inhibition using si-GAS5 prevented cellular apoptosis in H9C2 cells subjected to hypoxia-reoxygenation injury. Further, the silencing of GAS5 resulted in the upregulation of miR-532-5p, an interacting partner of GAS5, and reduced the expression of PTEN [164]. This study demonstrated that GAS5 may be a potential target for the attenuation of I/R injury [164]. A recent study showed that exosomes isolated from plasma samples of cardiac arrest/cardiopulmonary resuscitation (CA/CPR) patients contained abundant levels of GAS5, which targets inositol polyphosphate-4-phosphatase type II B (INPP4B) through the sponge effect with miR-137 [165]. MicroRNA array analysis showed a GAS5-miR-137-INPP4B network and the use of miR-137 mimic or silencing of GAS5 attenuated apoptosis of microglia [165]. Knockdown of GAS5 was also shown to reduce apoptosis and inflammation in cerebral ischemia/reperfusion injury through interaction with miR-26b-5p and the regulation of Smad1 expression [166]. The expression of GAS5 was upregulated following I/R injury, and treatment with RAPA significantly reduced GAS5 levels. These results suggest that GAS5 can be a viable target to alleviate cardiac apoptosis induced by MI and the improvement of cardiac function.

Go and KEGG pathway enrichment analysis offers an overview on the role of DEmiRs in various signaling mechanisms. Pathway enrichment analysis using GO and KEGG demonstrates the connection between gene expression and their functional pathways [167]. Our results show more than 125 miRs are involved in the regulation of cytoskeleton function, which plays a critical role in the maintenance and regulation of cardiomyocyte cell structure and integrity [168]. An event of MI triggers pathological signals that alter the balance between collagen synthesis and degradation and often leads to cardiac fibrosis and hypertrophy. Therefore, the maintenance of cytoskeletal homeostasis is critical for the normal function of the heart. Our data also demonstrated a significant number of miRs are involved cell cycle, apoptosis, cytokine–cytokine receptor interaction [169,170], and ErbB signaling [171].

Mitogen-activated protein kinases (MAPKs) are essential in different biological processes, i.e., cell survival, proliferation, differentiation, metabolism, and apoptosis. MAPK is associated with several pathological processes in cardiovascular disease, including hypertrophy, remodeling, and MI [172]. Several heart-specific miRNAs (miR-26a/b-5p, miR-27b-3p, miR-2392, and miR-3182) regulate MAPK signaling [173]. In the present study, pathway enrichment analysis also identified the involvement of more than 100 miRs in the MAPK signaling pathway. Zhu et al. revealed that the expression of lncRNA MALAT-1 was upregulated in coronary atherosclerotic heart disease (CAD) blood samples and endothelial progenitor cells [174], which could interact with miR-15b-5p and regulate MAPK and mTOR signaling. Another recent study showed an interaction between miR-214-3p and MAPK/mTOR signaling pathways in the atherosclerosis mice model [175].

Elevated levels of miR-21 in cardiofibroblasts activate the MAPK signaling pathway by inhibiting the sprouty homolog 1 (SPRY1) protein, which promotes cardiac remodeling and cardiac hypertrophy [176,177]. In addition, miR-19 could also alleviate fibrosis of CFs via attenuating the MAPKs pathway [178]. A recent study demonstrated that miRs (miR-125b-5p, miR-128-3p, and miR-30d-5p) present in small extracellular vesicles secreted by the brown adipose tissue provided exercise-induced cardioprotection against myocardial I/R injury through suppression of the MAPK pathway and by targeting Map3k5, Map2k7, and Map2k4 molecules [179]. Though the role of MAPK in cardiac physiology is well established, the regulation of MAPK signaling by miRNAs opens a new venue for research based on miRNA therapeutics.

## 5. Limitation of the Study

One of the limitations of this study is the use of the rat miRNA library for the miRArray chip analysis. Since the rabbit genome is poorly curated/annotated, we used the rat miRNA library chip for probing rabbit RNA samples. It should be noted that rat, mouse, human, and rabbit miRNAs are conserved and closely related to each other with a high percentage of homology in their nucleotide sequence, especially in the binding seed region (Supplement Figure).

## 6. Conclusions

We characterized DE miRs using an unbiased miRArray chip analysis and correlated with DE lncRNA after myocardial I/R injury in a conscious diabetic rabbit model. This study provides a global overview of DEmiRs in diabetes with and without MI. This study also provides insight into the novel molecular mechanism by which RAPA alters the miRNA signature. Our work highlights the complex network and interactions between noncoding RNAs involved in MI, like lncRNA, miRNA, and mRNA, and identifies novel targets for the development of potential RNA therapeutics for diabetic patients with MI.

## Figures and Tables

**Figure 1 cells-12-02820-f001:**
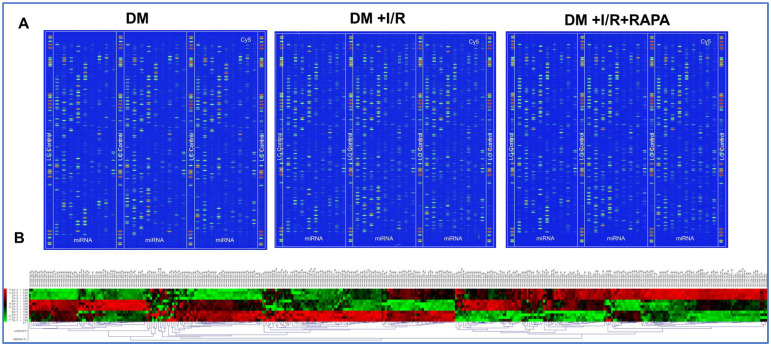
(**A**) Representative chip images and spots of miRNA array of diabetic rabbits (DM); diabetic rabbits following I/R (DM + I/R) and diabetic rabbits following I/R and treated with Rapamycin (DM + I/R + RAPA). (**B**) Hierarchical clustering plot (heat map) of all miRNAs with detectable intensity signal. Red color denotes high intensity miRNAs and green color indicates low signal miRNAs.

**Figure 2 cells-12-02820-f002:**
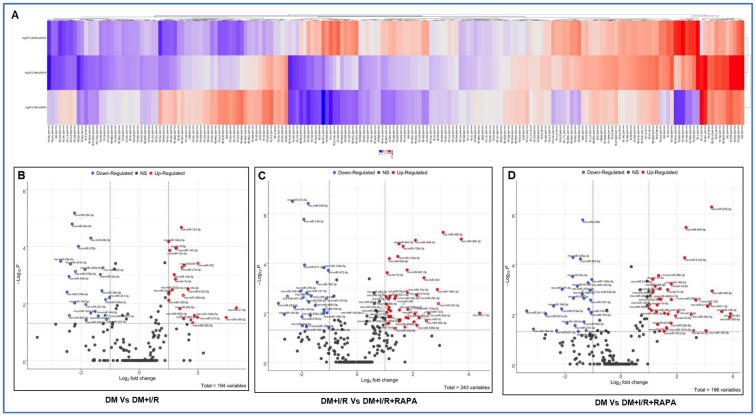
(**A**) Heat map of selected miRNAs that are differentially expressed in DM, DM + I/R, and DM + I/R + RAPA groups. Red color indicates upregulated miRs and blue color represents downregulated miRs. Volcano plot showing DEmiRs between (**B**) DM vs. DM + I/R, (**C**) DM + I/R vs. DM + I/R + RAPA, and (**D**) DM vs. DM + I/R + RAPA. *x*-axis represents log2 fold change and *y*-axis represents –log10 (adjusted *p*-value). Red dots represent upregulated DEmiRs and blue dots represent downregulated DEmiRs and black dots are miRs that are non-significantly altered. DEGs screening cutoff: fold change = 2 and adjusted *p*-value < 0.05.

**Figure 3 cells-12-02820-f003:**
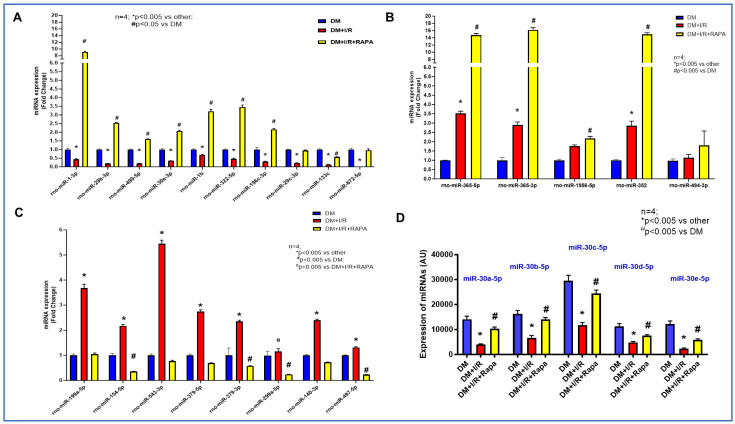
Differentially expressed miRNAs in DM + I/R and DM + I/R + RAPA as compared to DM. (**A**) Significantly upregulated miRs upon treatment with RAPA as compared to DM + I/R group, which were suppressed upon myocardial infarction (DM + I/R). (**B**) Significantly elevated miRs in DM + I/R group, which were further increased with RAPA treatment. (**C**) Negatively regulated miRs with RAPA treatment (DM + I/R + RAPA) that were significantly upregulated in DM + I/R compared to DM alone. (**D**) Expression profile of miR-30 family induced by RAPA treatment compared to DM + I/R and DM group.

**Figure 4 cells-12-02820-f004:**
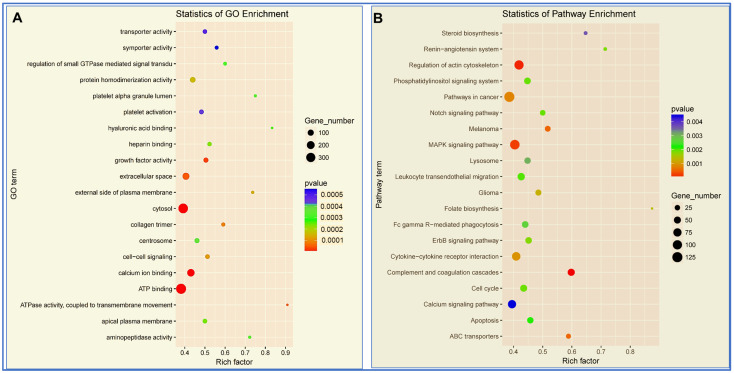
Functional pathway enrichment analysis of miRNAs using Gene Ontology (GO) and KEGG pathway based on number of genes. Enrichment score is equal to -log10 (*p*-value), which represents the significance level of GO and pathways. (**A**) Gene Ontology (GO) enrichment analysis of the cellular localization, biological processes, and molecular functions of the miRNAs. Large filled circle represents gene number of 300, medium filled circle denotes 200 miRs, and small filled circle shows genes that are 100 miRs. Red color represents *p* < 0.001, blue shows *p* < 0.005. (**B**) KEGG pathway analysis of significantly regulated miRs. Large filled circle represents gene number of 125, medium filled circle denotes 75 miRs, and small filled circle show genes that are 25 miRs. Red color represents a *p* < 0.001, blue shows *p* < 0.004.

**Figure 5 cells-12-02820-f005:**
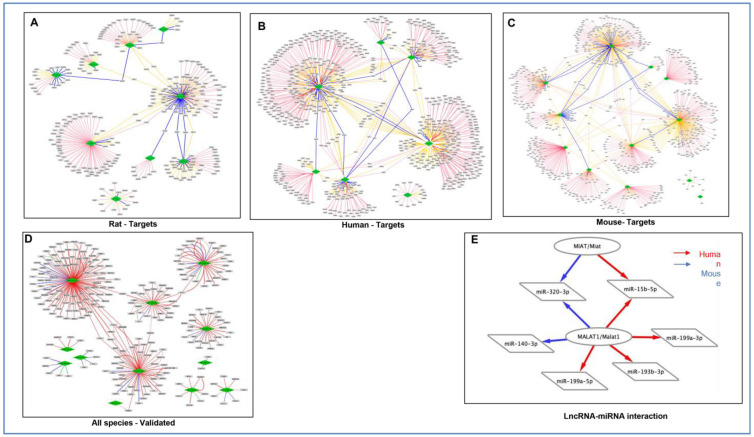
Interaction between DEmiRs and its mRNA target. (**A**) Rat, (**B**) Human, (**C**) Mouse, (**D**) All species—Validated. Red = one target; Orange = two targets; Blue = three targets. (**E**) Interaction network between lncRNA and miRNA. Prediction of interaction between lncRNAs MIAT with miR-320-3p, miR-140-3p, and lncRNA MALAT1 with miR-199a-3p, miR-15b-5p, and miR-193b-3p. Red = human, Blue = mouse, Green = rat; Straight line = 3’UTR; Dashed line = 5’UTR; Fish Bone line = CDS. Target Score ≥ 90 was used for prediction in miRDB interaction.

**Figure 6 cells-12-02820-f006:**
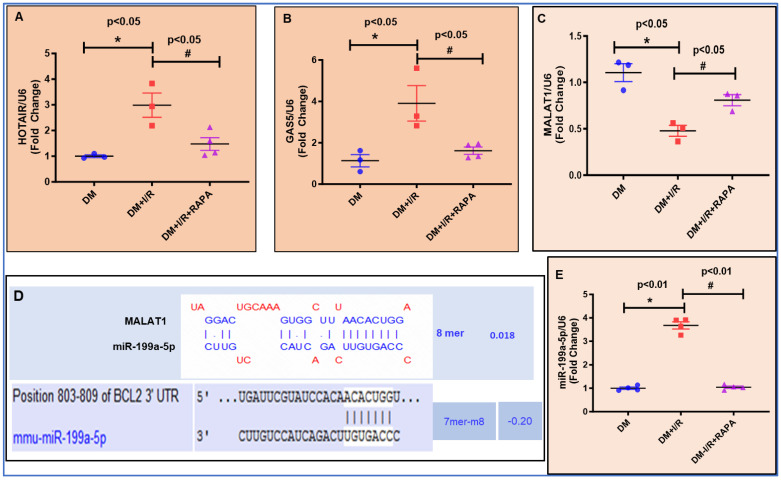
Expression analysis of lncRNA HOTAIR, GAS5, and MALAT1 using real-time PCR in DM, DM + I/R, and DM + I/R + RAPA groups. LncRNAs HOTAIR (**A**) and GAS5 (**B**) were upregulated upon myocardial infarction (DM + I/R) and significantly reduced with RAPA treatment (DM + I/R + RAPA). (**C**) MALAT1 expression was downregulated in DM + I/R and elevated by RAPA treatment (DM + I/R + RAPA). (**D**) Bioinformatics-based prediction to demonstrate the sponge effect of MALAT1 with miR-199a-5p and the binding sequence of miR-199a-5p with its mRNA target BCL2. (**E**) RNA expression of miR-199a-5p, an interacting partner of lncRNA MALAT1 in DM, DM + I/R, and DM + I/R + RAPA.

**Figure 7 cells-12-02820-f007:**
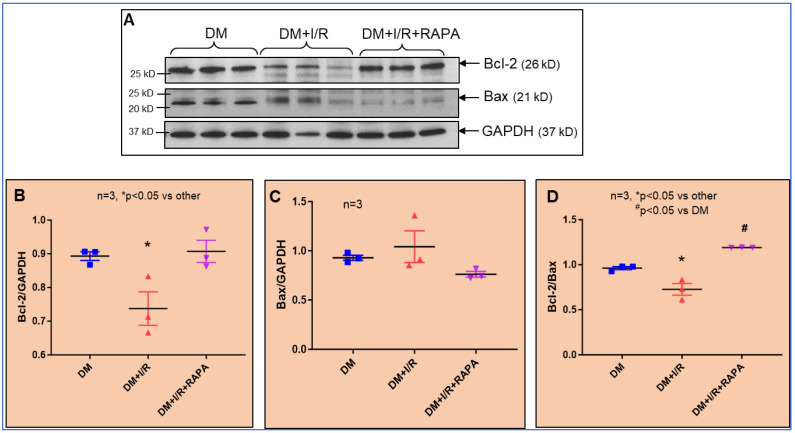
(**A**) Representative Western blots showing protein expression of BCL2 and Bax in DM, DM + I/R, and DM + I/R + RAPA groups from left ventricle tissue. (**B**) Densitometric analysis of the ratios of Bcl-2 and (**C**) Bax to GAPDH. (**D**) Ratio of Bcl-2 to Bax.

## Data Availability

The sequence data have been submitted to the BioSample database (hosted by the NCBI) (http://www.ncbi.nlm.nih.gov/biosample; accessed on 12 September 2023), under accession number PRJNA315318. R codes and related data/files for the study were deposited on GitHub at (https://github.com/AmyOlex/Samidurai-Das_mTOR_miRNA_Cardioprotection; accessed on 12 September 2023).

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
