# Peer review of "Integrated Analysis of lncRNA–miRNA–mRNA Regulatory Network in Rapamycin-Induced Cardioprotection against Ischemia/Reperfusion Injury in Diabetic Rabbits"

_cells, 2023, doi:10.3390/cells12242820_

Round 1
Reviewer 1 Report
Comments and Suggestions for Authors
The manuscript is very interesting. However, the following concerns should be addressed.
Immunoblots must be cropped in a way that retains information about antigen size and antibody specificity. The cropped images must retain sufficient area around the band(s) of interest, including the positions of at least one actual molecular weight marker above and one below the band(s). Please provide uncropped gels as supplementary material showing visible MW markers.
Bar graphs with error bars do not allow direct evaluation of the distribution of the data. The authors should present their data in scatter/dot plots (especially in case of a limited number of observations), showing the individual data points together with the average/error bars.
The discussion in its present form fails to interpret the data in the context of what is known in the field: it sounds somehow redundant, as it largely summarizes again data already presented in the Results without placing them in the proper scientific context.
The following pertinent topics should be mentioned/discussed:
doi: 10.3892/ijmm.2019.4407.
doi: 10.2217/pgs.13.143.
doi: 10.1038/srep07425
doi: 10.1161/CIRCRESAHA.119.315185.
doi: 10.3390/biom11020168.
Please avoid the use of greenish and reddish colors (issues in color blind Readers) in the same figure.
The font size in figure 2 is too small and barely readable: please split the 4 panels in 4 different figures.
The font size in figure 5 is too small and barely readable: please split the 5 panels in 5 different figures.
Comments on the Quality of English Language-
Author Response
- The manuscript is very interesting. However, the following concerns should be addressed.
Thank you for encouraging and insightful comments for revision of our paper. We have carefully incorporated all of suggestions in the revised version of the paper in track change mode. Following are the specific responses on each of the concerns:
- Immunoblots must be cropped in a way that retains information about antigen size and antibody specificity. The cropped images must retain sufficient area around the band(s) of interest, including the positions of at least one actual molecular weight marker above and one below the band(s). Please provide uncropped gels as supplementary material showing visible MW markers.
We have now provided WBs with visible MW markers. Also, we had already included the full western blot images with MW markers in the supplementary material.
- Bar graphs with error bars do not allow direct evaluation of the distribution of the data. The authors should present their data in scatter/dot plots (especially in case of a limited number of observations), showing the individual data points together with the average/error bars.
We have changed the bar graphs to scatter/dot plots for better visualization (Figure 6 A, B, C and E) and Figure 7 B, C and D.
- The discussion in its present form fails to interpret the data in the context of what is known in the field: it sounds somehow redundant, as it largely summarizes again data already presented in the Results without placing them in the proper scientific context.
We have revised the discussion by elaborating previous published studies, relevant to the focuses of the present manuscript.
- The following pertinent topics should be mentioned/discussed:
doi: 10.3892/ijmm.2019.4407.
We included the above reference in discussion section (Lines 615-621).
Gao et al, showed mTOR inhibition with RAPA reduced cardiomyocyte apoptosis fol-lowing MI stress[136]. Further, RAPA also inhibited the expression of cleaved caspase-3, promoted cardiomyocyte autophagy and alleviated ER stress through suppression of 78-kDa glucose-regulated protein (GRP78). Importantly, hyper-activation of mTOR sig-naling accelerates cardiac aging process by compromising autophagy and inhibition of mTOR activity rescued impaired autophagy in the heart, which delayed cardiac aging[135]. Treatment with RAPA improved contractile function and blocked the activa-tion of hypertrophic genes in aged mice heart. Moreover, RAPA also inhibits inflamma-tion associated with aging[137,138]
doi: 10.2217/pgs.13.143.
We cited this study (Santulli & Totary-Jain, Pharmacogenomics. 2013; PMID: 24024901) in the 2nd paragraph of the discussion section (lines 510-513)
doi: 10.1038/srep07425
We cited this reference and discussed about the role of mTOR in cardiac aging (PMID: 25502776)” in line 618-623).
doi: 10.1161/CIRCRESAHA.119.315185.
We cited the above reference in the Discussion section ( Ref# 120; Boada et al. Cir Res. 2020; lines 573-576) with the following statements:
‘Targeted delivery of RAPA via biomimetic nanoparticles for 7 days decreased macro-phage proliferation in the aorta with the reduction of key pro-inflammatory cytokines in a murine model of atherosclerosis (PMID: 31647755)’.
doi: 10.3390/biom11020168.
Included the above reference in the discussion section (Daneshgar et al. Biomolecules 2021) as well as Dai et al. Aging Cell 2014 (Lines 618-623) to elaborate the beneficial effect of rapamycin treatment against aging-associated disorders.
- Please avoid the use of greenish and reddish colors (issues in color blind Readers) in the same figure.
We have replaced the figure with blue and yellow colors. To be more specific, Figure 3, 6 and 7 have new color pattern of blue for DM, red for DM+I/R and yellow for DM+I/R+RAPA. Unfortunately, changing the color in figures obtained using extensive coding and bioinformatic tools will be cumbersome. We have noted this point and will avoid using green and red color in the same figure in our future publications.
- The font size in figure 2 is too small and barely readable: please split the 4 panels in 4 different figures.
We understand that it is difficult to read the font in figure 2. Unfortunately due to the restriction on the number of figures in the article by the journal, we are unable to split the figures and provide it in an individual panel. However, a high-resolution image of figure 2 and all relevant images from bioinformatic analysis are available from the link provided in the section 2.12. Data Availability (https://github.com/AmyOlex/Samidurai-Das_mTOR_miRNA_Cardioprotection2.14).
The font size in figure 5 is too small and barely readable: please split the 5 panels in 5 different figures.
A high-resolution image of figure 5 is available from the link provided in the section 2.12. Data Availability (https://github.com/AmyOlex/Samidurai-Das_mTOR_miRNA_Cardioprotection2.14).
Reviewer 2 Report
Comments and Suggestions for Authors
Dear Author,
The manuscript
"Integrated Analysis of lncRNA-miRNA-mRNA Regulatory Network in Rapamycin-Induced Cardioprotection Against Ischemia/Reperfusion Injury in Diabetic Rabbits"
is an interesting study I however I feel if the authors address the following comments that can improve the overall presentation of study.
1. Abstract need a more scientific approach rather supplying generic information.
2. Revise keywords of the article.
3. In Manuscripts, there are several grammatical and spelling mistakes author need to carefully remove mistakes.
4. Line 33. Could you provide a brief overview of the mechanisms through which diabetes augments the deleterious outcomes following myocardial infarction (MI)? This could help establish a clearer link between DM and MI.
5. Line 34-37. While the statistics for diabetes prevalence in the US and globally are mentioned, it might be helpful to briefly discuss the potential reasons behind this significant increase. Is this primarily due to lifestyle factors, improved diagnostic methods, or other factors?
6. Line 45: How specifically does diabetes alter the epigenetic pattern and molecular mechanisms of gene expression? Any specific pathways or markers mentioned in the literature?
7. Line 59: Could you provide examples of tissue-specific lncRNAs and their functions in cellular regulation?
8. Line 70: Could you elaborate on the specific epigenetic controls that result from the three-way interaction between lncRNA-miRNA-mRNA?
9. Line 102: It is mentioned that the Institutional Animal Care and Use Committee (IACUC) protocol number is AM10109. Could you provide a brief description or reference for this protocol for better context?
10. Line 115: How was anesthesia administered during the implantation of the balloon occluder? Were there any specific anesthetics used?
11. Line 128: Could you briefly explain what parameters were used to verify the concentration and purity of the isolated RNA? Additionally, if any specific thresholds were applied, please provide those details.
12. Lines 134-140: Could you clarify the specifics of the oligonucleotide tag ligation and fluorescent dye staining process? Were there any specific controls or standards used in this process?
13. Lines 143-145: Provide more information about the poly(A) tailing process, including any specific reagents or conditions used.
14. Line 147-152: Were there any specific conditions or temperatures during the overnight hybridization?
15. Line 154-159: It's mentioned that miRNA normalization and differential expression analysis were performed by LC Sciences. Could you elaborate on the specific methods they used for normalization and how they handled technical replicates?
16. Line 161-169: Could you clarify what criteria were used for selecting miRNAs for the heatmap and volcano plot? Were there any specific R-Studio packages or functions used for generating these visualizations?
17. Line 173: When using the miRDB database for miRNA-mRNA target prediction, did you consider any specific parameters for filtering the predictions?
18. Line 252: It would be helpful to specify the specific miRNAs that are differentially expressed. Could you provide a list or mention some examples?
19. Line 262: Could you provide some additional context on why these specific miRNAs were chosen for analysis? Is there existing literature supporting their relevance in this context?
20. Line 290-292: For the miRNAs that were increased by more than 2-fold in post-I/R heart after RAPA treatment, could you discuss their potential significance in the context of diabetic myocardium?
21. Regarding Figure 3B, where miR-365-5p, miR-365-3p, miR-352, miR-494-3p, and miR-1956-5p were upregulated post I/R injury in diabetic rabbits, followed by further induction with RAPA treatment, can you provide statistical significance levels for these changes?
22. Line 440: Could you clarify what you mean by "resolve the complexity associated with gene expression"?
23. Line 460: Could you provide more insight into how the global change in miRNA profile with RAPA treatment correlates with the reduction in apoptosis, as observed in your previous study?
24. Line 662: The regulation of MAPK signaling by miRNAs is an intriguing point. Have there been any specific miRNAs identified in this study that play a significant role in regulating MAPK signaling in the context of cardiovascular disease?
The similarity index is bit to high, please reduce that.
best regards,
Comments on the Quality of English Language
Minor editing is needed before fin al publication
Author Response
The manuscript "Integrated Analysis of lncRNA-miRNA-mRNA Regulatory Network in Rapamycin-Induced Cardioprotection Against Ischemia/Reperfusion Injury in Diabetic Rabbits" is an interesting study I however I feel if the authors address the following comments that can improve the overall presentation of study?
Thank you for the encouraging comments. Following are the responses to each of the concern.
- Abstract need a more scientific approach rather supplying generic information.
We revised the abstract with briefing our results.
- Revise keywords of the article.
We revised the keywords.
- In Manuscripts, there are several grammatical and spelling mistakes author need to carefully remove mistakes.
We have revised the entire manuscript and corrected the errors.
- Line 33. Could you provide a brief overview of the mechanisms through which diabetes augments the deleterious outcomes following myocardial infarction (MI)? This could help establish a clearer link between DM and MI.
We provided a brief overview of the mechanisms associated in diabetes-induced myocardial infarction (lines 43-57).
The mortality rate after an incident of MI is higher in diabetes patients compared to non-diabetic subjects [3-5]. In diabetic patients, hyperglycemia, insulin resistance with a compensatory hyperinsulinemia, excessive production of fatty acid and imbalanced lipid metabolism lead to an increase in systemic oxidative stress, protein kinase C activation, and the production of advanced glycation product (AGE)[6,7]. These all promote endothe-lial cell apoptosis and impair endothelial dysfunction, which causes vascular inflamma-tion and vasoconstriction[8]. Moreover, the higher tendency of coronary artery calcifica-tion (CAC) in diabetic patients, due to increased oxidative stress and inflammatory cyto-kine production, endothelial dysfunction, alteration in mineral metabolism and release of osteoprogenitor cells from the marrow into the circulation lead to an increase in arterial rigidity and atherosclerotic plaque burden[9]. The presence of diabetic neuropathy also enhances the incidence of death after an episode of MI [10] largely due to the lack of pain and attention while experiencing MI.
- Line 34-37. While the statistics for diabetes prevalence in the US and globally are mentioned, it might be helpful to briefly discuss the potential reasons behind this significant increase. Is this primarily due to lifestyle factors, improved diagnostic methods, or other factors?
We have elaborated on the potential reasons for increase rate of diabetes in the 1st paragraph of Introduction (Lines 59-64) as described in the following text.
“Age adjusted diabetes rate is expected to increase by 59.7% between the year 2021 and 2050, resulting in a global population of 1·31 billion people living with diabetes in 2050 (PMID: 37356446). The alarming increase in the rate of diabetes is largely due to prevalence of childhood obesity and the presence of gestational diabetes. There are several factors that increase the rate of diabetes including lack of exercise, lifestyle changes, awareness, diet (PMID: 25178569); and demographic characteristics, (PMID: 36960397)”.
- Line 45: How specifically does diabetes alter the epigenetic pattern and molecular mechanisms of gene expression? Any specific pathways or markers mentioned in the literature?
We have added the following discussion to address the link between diabetes and epigenetics in the 2nd paragraph of the Introduction (Lines 72-80).
“Differentially methylated regions (DMRs) in the whole genome especially pancreatic and duodenal homeobox 1 (PDX1), a key transcription factor present in the pancreatic islet cells that regulate the insulin secretion (PMID: 28052964), has been reported to play an important role in diabetes. Several studies also demonstrate that diabetes influence epigenetics and alter the expression of gene pattern. Human islets incubated in high glucose (19 mM) altered the methylation pattern of five key genes such as GLRA1, RASD1, VAC14, SLCO5A1 and CHRNA5 (PMID: 29183809). Gene enrichment also suggest that signaling pathways including TGF-beta, Notch and SNARE interactions in vesicular transport are involved in the islet function and diabetes”.
- Line 59: Could you provide examples of tissue-specific lncRNAs and their functions in cellular regulation?
We provided following examples for tissue-specific lncRNAs and their specific regulatory roles in different tissues in the 2nd paragraph of Introduction (lines 93-102).
“LncRNAs can be identified as ubiquitously expressed long non-coding RNAs (UE lncRNAs) from tissue-specific lncRNAs (TS lncRNAs) (PMID: 26760768). For example, the expression of lncRNA MIR17HG is specific to lungs, which plays an important role in cell survival, proliferation, differentiation, and angiogenesis (PMID: 17765889). MIR17HG level is less abundant in non-small cell lung cancer and overexpression of MIR17HG reduces the methylation of miR-142-3p gene to upregulate its expression, which inhibit cancer cell invasion and migration (PMID: 32228546). Several TS lncRNAs such as MYH2, TNNT2 and SLC8A1 are unique to heart tissue and have known function in muscle contraction and relaxation, atrial cardiac muscle tissue morphogenesis and cardiac muscle development”.
- Line 70: Could you elaborate on the specific epigenetic controls that result from the three-way interaction between lncRNA-miRNA-mRNA?
We emphasized the epigenetic regulation and three-way interaction between lncRNA-miRNA-mRNA in lines 119-128 of the Introduction section as described in the following text.
“Non-coding RNAs including lncRNA and miRNA through their sponge mechanism can regulate the expression of mRNA and also dictate the DNA methylation and histone modification (PMID: 27841002). One of the classical examples for the three-way interaction between lncRNA-miRNA-mRNA is evident from the report by Cheng et al, where lncRNA HOTAIR epigenetically suppressed the expression of miR-122 through DNA methyltransferases (DNMTs)-mediated DNA methylation via upregulation of Enhancer of zeste homolog 2 (EZH2) (PMID: 30195653). The suppression of miR-122 by HOTAIR through DNA methylation resulted in the activation of Cyclin G1 and enhanced tumor growth in hepatocellular carcinoma (HCC)”.
- Line 102: It is mentioned that the Institutional Animal Care and Use Committee (IACUC) protocol number is AM10109. Could you provide a brief description or reference for this protocol for better context?
A detailed protocol and procedures followed for the animal experiments is under the protocol number is AM10109 of Institutional Animal Care and Use Committee at the Virginia Commonwealth University, which has been described in our previous publication. PMID: 33319180 and PMID: 34485939.
- Line 115: How was anesthesia administered during the implantation of the balloon occluder? Were there any specific anesthetics used?
The rabbits were anesthetized by intramuscular (i.m.) administration of ketamine (35 mg/kg), Xylazine (5 mg/kg) and Atropine (5 mg/kg) during the implantation of the balloon occluder. A detailed procedure followed in the Conscious I/R injury rabbit model including the methods involved in the Implantation of balloon occluder can be found in our previous publication. PMID: 33319180 and PMID: 34485939
- Line 128: Could you briefly explain what parameters were used to verify the concentration and purity of the isolated RNA? Additionally, if any specific thresholds were applied, please provide those details.
The concentration and the purity of the isolated RNA was verified using Nanodrop ND-1000 spectrophotometer (Agilent technologies, CA, USA). The ratio of absorbance at 260 nm and 280 nm is used to assess the purity of RNA. A ratio of A260/A280 between 1.8 and 2.0 was considered as acceptable purity and the impurities was measured by a ratio of A260/A230, where a value range of 1.8 to 2.0 was accepted as less impure.
- Lines 134-140: Could you clarify the specifics of the oligonucleotide tag ligation and fluorescent dye staining process? Were there any specific controls or standards used in this process?
This procedure was done commercially at LC Sciences Company, Houston, Texas. The oligonucleotide adapter sequence contains a tag segment for capturing fluorescent dye during a later dye staining process. Array also contains control probes for chip quality analysis, spiking-in controls, internal positive controls, and sometimes negative controls as well. More than 12 spike-in controls are used in each assay.
After RNA hybridization, tag-conjugating Cy3 dye was circulated through the microfluidic chip for dye staining. Fluorescence images were collected using a laser scanner (GenePix 4000B, Molecular Device) and digitized using Array-Pro image analysis software (Media Cybernetics). For a dual sample assay, Cy3 and Cy5 reaction solutions were used separately for the two corresponding RNA samples.
- Lines 143-145: Provide more information about the poly(A) tailing process, including any specific reagents or conditions used.
Microarray assay was performed by LC Sciences. A strand-labeling reaction was carried out with total RNA (1 ug), control oligo-nucleotides, App-Cap3(5’-adenylated and 3’-dideoxy oligo-adaptor, Integrated DNA Technologies) and T4 RNA ligase (Rnl2tr-K227Q, New England BioLabs) in T4 RNA ligase buffer with PEG 8000 and RNase inhibitor (Promega). The reaction mixture was incubated at 16ºC for 16 hours and stopped by adding equal volume of hybridization buffer.
- Line 147-152: Were there any specific conditions or temperatures during the overnight hybridization?
A detailed procedure of the miRNA-Array assay can be found in the publication by Zhou et al. [78]. Hybridization was performed at 40°C for 16 hours on a µParaflo microfluidic chip using a micro-circulation pump (Atactic Technologies). On the microfluidic chip, each detection probe consisted of a chemically modified nucleotide coding segment complementary to target microRNA (from miRBase, http://mirbase.org) or other RNA (control or customer defined sequences) and a spacer segment of polyethylene glycol to extend the coding segment away from the substrate.
- Line 154-159: It's mentioned that miRNA normalization and differential expression analysis were performed by LC Sciences. Could you elaborate on the specific methods they used for normalization and how they handled technical replicates?
Normalization was carried out using LOWESS (locally weighted scatterplot smoothing) method on background-subtracted data (PMID: 28018492; PMID: 12538238).The normalization removes system related variations, such as sample amount variations, labeling dye differences, signal gain differences of scanners, and nonlinear signal intensity response to target concentration so that biological variations are faithfully revealed. Standard internal positive controls include 5 to 6 probes targeting at different sections of 5S ribosomal RNA of corresponding species. Most probes are repeated three times or more.
- Line 161-169: Could you clarify what criteria were used for selecting miRNAs for the heatmap and volcano plot? Were there any specific R-Studio packages or functions used for generating these visualizations?
Differentially expressed miRNAs with an expression level greater than 2-fold change was included in the heatmap and volcano plot. Heatmap was generated using R studio software (Heatmap3 package) and volcano plot was constructed using R studio-Bioconductor package. The details stated above are included in the methods section under the subtitle “2.6. Heat Map and Volcano Plot”.
- Line 173: When using the miRDB database for miRNA-mRNA target prediction, did you consider any specific parameters for filtering the predictions?
The miRNA-mRNA target prediction was performed by setting a filter parameter target Score >=90 for their miRDB interaction. This is detailed in the methods section “2.7. miRNA-mRNA Target Prediction”. We also added the note about the filter parameter in the legend of figure 5.
- Line 252: It would be helpful to specify the specific miRNAs that are differentially expressed. Could you provide a list or mention some examples?
The list of important miRNAs that are differentially expressed are listed in the results section “3.2. Heat Map and Volcano plot and 3.3. Rapamycin Alters the Expression of miRNAs”
miR-1-3p, miR-1b, miR-29b-3p, miR-29c-3p, miR-30e-3p, miR-133c, miR-196c-3p, miR-322-5p, miR-499-5p and miR-672-5p,
The following miRNAs were upregulated by RAPA (Figure 2 C)
miR-1-3p, miR-29b-3p, miR-29c-3p, miR-499-5p, miR-365-3p, miR-1956-5p, miR-30e-3p, miR-1b, miR-322-5p, miR-196c-3p, miR-352, miR-494-3p and miR-133c
The following miRNAs were downregulated by RAPA
miR-672-5p, miR-539-5p, miR-154-5p, miR-543-3p, miR-379-3p, miR-379-5p, miR-299a-5p, miR-140-3p, miR-140-5p miR-497-5p, miR-379-3p, miR-411-3p, miR-199a-5p, miR-34a-5p and miR-495
- Line 262: Could you provide some additional context on why these specific miRNAs were chosen for analysis? Is there existing literature supporting their relevance in this context?
We performed an unbiased global miRNA array profile in our study and selected few specific miRNAs for further analysis that were differentially and significantly regulated between groups by a fold change greater than 2.
- Line 290-292: For the miRNAs that were increased by more than 2-fold in post-I/R heart after RAPA treatment, could you discuss their potential significance in the context of diabetic myocardium?
We emphasized the significances of miR-1, miR-29, miR-499, miR-133 in diabetic myocardium in the respective sections under discussion.
- Regarding Figure 3B, where miR-365-5p, miR-365-3p, miR-352, miR-494-3p, and miR-1956-5p were upregulated post I/R injury in diabetic rabbits, followed by further induction with RAPA treatment, can you provide statistical significance levels for these changes?
We apologize for not including the statistical details for figure 3b. We have added the statistical significance for Figure 3B in our revised version.
- Line 440: Could you clarify what you mean by "resolve the complexity associated with gene expression"?
Gene expression is regulated by several transcription factors and coordinated through key players such as non-coding RNAs. Deciphering the role of specific ncRNA or the interaction among lncRNA and miRNA can unwind the complex knots involved in gene expression in response to cell stimuli.
- Line 460: Could you provide more insight into how the global change in miRNA profile with RAPA treatment correlates with the reduction in apoptosis, as observed in your previous study?
We previously demonstrated that RAPA protects heart of diabetic rabbit against I/R injury by reducing apoptosis through miR-302a-PTEN-AKT pathway [51]. We noticed that mTORC1 activation (indicated by phosphorylation of ribosomal S6) was increased, whereas mTORC2 activation (indicated by phosphorylation of AKT and miR-302a with induction of PTEN (the target of miR-302a) were reduced following I/R injury in diabetic heart. RAPA treatment at reperfusion suppressed mTORC1 activity and PTEN, with res-toration of miR-302a level and mTORC2 activity.
In the present study, GO and KEGG enrichment analysis presented in figure 4 shows more than 25 miRNAs are involved in the apoptosis process with a statistical significance p value of 0.002. We explained the correlation of miR-1, miR-29b-3p, miR-499 and miR-133 with reduction of apoptosis after RAPA treatment.
- miR-1 overexpression also inhibited cardiac fibrosis and apoptosis by reducing the mRNA expression of transforming growth factor beta‐1 (Tgfb1) and increasing protein expression of Bcl‐2, while reducing the level of Bax, thereby altering Bcl-2/Bax ratio [103].
- Several other studies also demonstrated the therapeutic effect of miR-29b in diabetes, cardiac fibrosis, apoptosis [110] and MI [111,112].
- AgomiR overexpression of miR-499-5p in acute MI model limited infarct size and reduced cardiomyocyte apoptosis through targeting programmed cell death 4 (PDCD4) [114]. Other studies also support the notion that miR-499 targets PDCD4 and strongly suggested that overexpression of miR-499 is beneficial and cardioprotective through reducing apoptosis [115-117]. It was also shown that miR-499 binds to SOX6 and reduced cardiomyocyte apoptosis [118].
- Line 662: The regulation of MAPK signaling by miRNAs is an intriguing point. Have there been any specific miRNAs identified in this study that play a significant role in regulating MAPK signaling in the context of cardiovascular disease?
Thank you for raising this important question. We elaborated the role of MAPK in cardiovascular diseases and its regulation with specific miRNAs.
Mitogen-activated protein kinases (MAPKs) are essential in different biological processes, i.e., cell survival, proliferation, differentiation, metabolism, and apoptosis. Several heart-specific miRNAs (miR-26a/b-5p, miR-27b-3p, miR-2392 and miR-3182) regulate MAPK signaling (PMID: 37393060). In the present study, GO and KEGG enrichment analysis presented in figure 4 shows more than 100 miRs are involved in MAPK signaling pathway.
Zhu et al. revealed that the expression of lncRNA MALAT-1 was upregulated in coronary atherosclerotic heart disease (CAD) blood samples and endothelial progenitor cells (PMID: 30787203), which could interact with miR-15b-5p and regulate MAPK and mTOR signaling. Another recent study showed an interaction between miR-214-3p and MAPK/mTOR signaling pathways in atherosclerosis (PMID: 37778998).
Elevated level of miR-21 in cardiofibroblast activates the MAPK signaling pathway by inhibiting sprouty homolog 1 (SPRY1) protein, which promotes cardiac remodeling and cardiac hypertrophy (PMID: 19043405; PMID: 24804616). In addition, miR-19 could also alleviate fibrosis of CFs via attenuating the MAPKs pathway (PMID: 36539014).
Also, a recent study demonstrated that miRs (miR-125b-5p, miR-128-3p, and miR-30d-5p) present in small extracellular vesicles secreted by the brown adipose tissue provided exercise-induced cardioprotection against myocardial ischemia/reperfusion injury through suppression of the MAPK pathway and by targeting Map3k5, Map2k7, and Map2k4 molecules (PMID: 35387487).
The similarity index is bit to high, please reduce that.
We tried our best to rewrite the manuscript to avoid high similarity.
Round 2
Reviewer 1 Report
Comments and Suggestions for Authors
-
Comments on the Quality of English Language-